# Three-Dimensional Envelope and Subunit Interactions of the Plastid-Encoded RNA Polymerase from *Sinapis alba*

**DOI:** 10.3390/ijms23179922

**Published:** 2022-08-31

**Authors:** Rémi Ruedas, Soumiya Sankari Muthukumar, Sylvie Kieffer-Jaquinod, François-Xavier Gillet, Daphna Fenel, Grégory Effantin, Thomas Pfannschmidt, Yohann Couté, Robert Blanvillain, David Cobessi

**Affiliations:** 1CNRS, CEA, IBS, University Grenoble Alpes, 38000 Grenoble, France; 2CNRS, CEA, INRAE, IRIG-LPCV, University Grenoble-Alpes, 38000 Grenoble, France; 3INSERM, CEA, University Grenoble Alpes, UMR BioSanté U1292, CNRS, CEA, FR2048, 38000 Grenoble, France

**Keywords:** *Sinapis alba*, plastid-encoded RNA polymerase, PEP associated proteins, transcription, photomorphogenesis, photosynthesis, chloroplast biogenesis

## Abstract

RNA polymerases (RNAPs) are found in all living organisms. In the chloroplasts, the plastid-encoded RNA polymerase (PEP) is a prokaryotic-type multimeric RNAP involved in the selective transcription of the plastid genome. One of its active states requires the assembly of nuclear-encoded PEP-Associated Proteins (PAPs) on the catalytic core, producing a complex of more than 900 kDa, regarded as essential for chloroplast biogenesis. In this study, sequence alignments of the catalytic core subunits across various chloroplasts of the green lineage and prokaryotes combined with structural data show that variations are observed at the surface of the core, whereas internal amino acids associated with the catalytic activity are conserved. A purification procedure compatible with a structural analysis was used to enrich the native PEP from *Sinapis alba* chloroplasts. A mass spectrometry (MS)-based proteomic analysis revealed the core components, the PAPs and additional proteins, such as FLN2 and pTAC18. MS coupled with crosslinking (XL-MS) provided the initial structural information in the form of protein clusters, highlighting the relative position of some subunits with the surfaces of their interactions. Using negative stain electron microscopy, the PEP three-dimensional envelope was calculated. Particles classification shows that the protrusions are very well-conserved, offering a framework for the future positioning of all the PAPs. Overall, the results show that PEP-associated proteins are firmly and specifically associated with the catalytic core, giving to the plastid transcriptional complex a singular structure compared to other RNAPs.

## 1. Introduction

DNA-dependent RNA polymerases (RNAPs) are central enzymes of gene expression, which transcribe the genetic information encoded in DNA into single-stranded RNAs, some of which are suitable for translation. RNAPs exist in highly varying degrees of complexity ranging from single subunit enzymes in T3/T7 phages to highly multimeric enzymes in eukaryotes. Eubacterial multimeric RNAPs share a common catalytic core composed of two large subunits called β and β’, a dimer of α subunits and a monomer of the ω subunit [1,2,3]. For specific transcriptional activity, RNAPs require additional proteins such as σ factors that mediate the recognition of gene promoters and are essential to initiate transcription. The three-dimensional structures of RNAPs have been solved for eukaryotic and prokaryotic RNAPs in several states [3,4,5]. Structural comparisons of RNAPs have shown that, even when the sequence identity is low, the overall shape of the five core subunits is largely conserved [3]. Furthermore, homologous regions at the structural level have been identified between the bacterial and eukaryotic RNAPs, suggesting that the fold is better conserved than the amino acid sequences. The essential residues and regions for effective transcription are, however, conserved, indicating that the enzymes share a common transcription mechanism [1]. In eukaryotes, several RNAPs are involved in the transcription of nuclear genes (RNAPs I, II and III), while a specific phage-type RNAP transcribes the mitochondrial DNA. Plant cells, in addition, possess a third genome in plastids with complex transcriptional machinery to express it. Plastids evolved from the engulfment of an ancient cyanobacterium into a mitochondriate proto-eukaryote around 1.5 billion years ago [6]. Thereafter, a massive lateral transfer of cyanobacterial genes into the nucleus reshaped the two genomes [7]. As a result, most plastome (chloroplast DNA, cpDNA) of today’s plastids contains only about 120 genes [8], encoding (i) components of the plastid gene expression machinery (the core subunits of the prokaryotic-type RNA polymerase, ribosomal proteins, tRNAs and rRNAs); (ii) subunits of each of the major functional photosynthesis-related complex (e.g., ribulose-1,5-bisphosphate carboxylase/oxygenase (Rubisco), photosystem I and II (PSI and PSII), cytochrome b6f complex, NADH dehydrogenase and the ATP synthase) and (iii) a few proteins involved in other essential processes, such as protein import, fatty acid synthesis or protein homeostasis (e.g., YCF1 and 2, AccD and ClpP1) [9,10]. Despite the limited number of plastid genes, chloroplasts contain 2500–3500 different proteins [11]; thus, the vast majority of chloroplast proteins are encoded by the nuclear genome and must be post-translationally imported. The expression of the cpDNA is, however, essential to chloroplast biogenesis and functions since drug-based or genetic impairments of plastid gene expression result in albinism [12].

Transcription of the plastome involves a single-subunit nuclear-encoded T3/T7 phage-type RNA polymerase (NEP) and the multi-subunit plastid-encoded prokaryotic-type RNA polymerase (PEP). Briefly, the NEP enzyme transcribes the so-called ‘house-keeping’ genes (including rpo genes encoding the core subunits of the PEP), while the PEP preferentially transcribes genes encoding proteins of the photosynthetic complexes, as well as tRNA genes [13,14]. However, some plastid genes possess promoters for NEP and PEP so that they can be transcribed by both RNA polymerases [15]. Furthermore, the division of labor between the two RNA polymerases changes with the developmental stage, and a clear-cut separation between NEP and PEP transcribed genes remains difficult [16]. The catalytic core enzyme of PEP comprises four subunits called α, β, β’ and β”, encoded by the genes rpoA, rpoB, rpoC1 and rpoC2, respectively [17,18]. Biochemical studies performed in dark-grown mustard revealed that the core subunits assemble to form the prokaryotic-like enzyme PEP-B [19,20,21]. In angiosperm, seedlings illumination initiates a light signaling cascade that triggers photomorphogenesis and chloroplast biogenesis. This involves a structural reorganization of the PEP-B enzyme by association, with additional subunits resulting in a much larger multi-subunit PEP-A complex. Biochemical purifications performed in several plants revealed that the complex comprises at least 16 different proteins with an overall molecular mass of more than 900 kDa [22,23]. MS analyses of the mustard PEP-A complex allowed the identification of 10 PEP-associated proteins (PAPs) that are stably bound to the complex. Two additional proteins (PAP11/MurE and PAP12/pTAC7) were then added to the list of PAPs according to a set of criteria, including biochemistry (presence in the complex) and genetics (albino syndrome of the mutant) [18]. These PAPs are all encoded by the nuclear genome and must be imported in the stroma from the cytosol. The genetic inactivation of any of these 12 PAPs causes a severe block or disturbance of chloroplast biogenesis, indicating that the reorganization of the PEP complex represents a critical step in chloroplast biogenesis [12,23,24,25,26,27,28,29,30,31]. Therefore, understanding chloroplast biogenesis associated with photosynthesis in angiosperms requires studying the nuclear-encoded PAPs that, added to PEP, regulate gene expression while protecting the machinery from the threatening reactions of photosynthesis.

In contrast to RNAPs I, II and III, for which several three-dimensional structures were solved, the PEP-A structure remains unknown. Based on sequence homology, it is assumed that the PEP core enzyme would resemble that of the bacterial RNA polymerase (bRNAP). With the exception of PAP9, whose 3D structure was recently solved [32], only structure predictions of PAPs have been calculated based on their amino acid sequences, searching structural databases for homologous domains [18].

Here, we report the characterization of the PEP complex purified from *S. alba* cotyledons. A MS-based proteomic analysis identified all known PEP subunits and additional members, such as FLN2 and pTAC18. A chemical crosslinking coupled to MS approaches highlighted some interacting peptides in the PEP complex and provided initial structural information in the form of protein clusters, highlighting the relative position of some subunits with their surfaces of interaction. Using negative stain electron microscopy, we calculated the first 3D envelope of the PEP-A complex, showing together with the MS analyses that the PAPs are firmly and reproducibly associated with the catalytic core, each likely at its specific site. Interestingly, some surfaces of the interactions between the core and PAPs correspond to conserved regions of PAP-containing clades that are otherwise variable when bRNAPs are also considered.

## 2. Results

### 2.1. The PEP Complex and Its Associated Proteins

We used a MS-based label-free quantitative proteomic analysis to characterize the *S. alba* PEP-enriched fraction isolated from the chloroplasts of mustard cotyledon. An established purification scheme was used with slight modifications [33] (Figure 1a).

**Figure 1 ijms-23-09922-f001:**
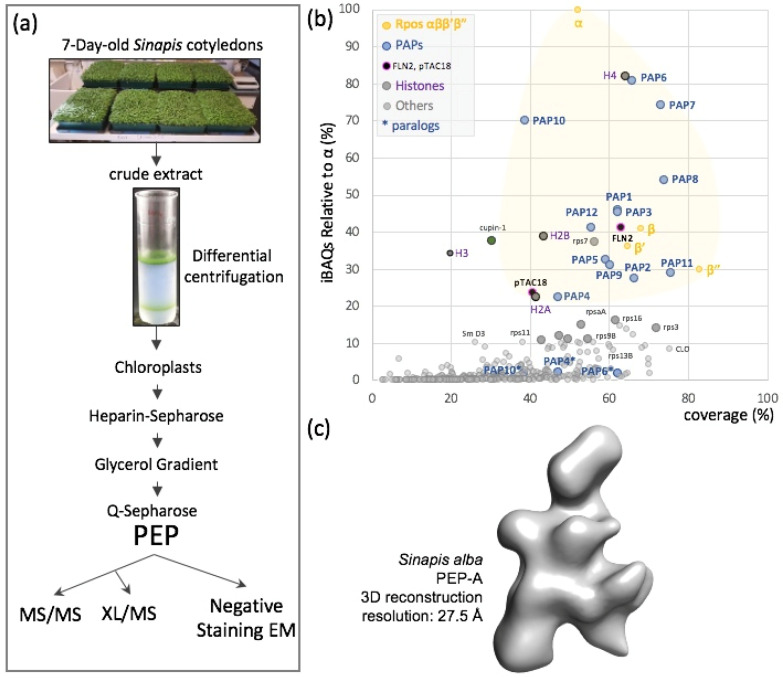
PEP composition and three-dimensional envelope. (**a**) Organelle fractionation, purification scheme and sample processing for mass spectrometry (MS/MS) or crosslinking mass spectrometry (XL/MS) or negative staining electron microscopy (eM). (**b**) Mass spectrometry data presented as relative iBAQ values to that of α (iBAQr) as a function of the corresponding protein coverage expressed in percentage. Subunits α, β, β’ and β” are in yellow, PAPs in blue, suspected permanent residents in black, histones in magenta and suspected purification contaminants in different shades of grey. In the shaded yellow area fall all the expected components of the PEP-A complex and correspond to the major protein mass contribution to the purified sample. (**c**) *Sinapis alba* PEP-A envelope calculated from negative staining EM acquisitions (see Figure 2 for details).

More than 400 different proteins were reproducibly identified and quantified in three independent preparations of PEP (Appendix A). Their relative abundances within the PEP fraction were approximated using their extracted iBAQ values [34], showing that these proteins were distributed over four orders of magnitude. Among the 24 most abundant proteins, representing ~60% of the total amounts of proteins within the fraction, we identified the four core subunits (α, β, β’ and β”) and the twelve PAPs (Figure 1b and Appendix A). The α subunit was found to be approximately twice more abundant than the β subunit, consistent with a stoichiometry of two α subunits per one β subunit in the catalytic core complex, as described in eubacterial RNAPs. Besides the 16 known PEP subunits, we identified PAP6/FLN1 paralogous fructokinase-like protein 2 (FLN2) and pTAC18 that were identified as a subunit of the plastid transcriptionally active chromosome [35]. The shortlist also contains two unexpected proteins, one homologous of the *A. thaliana* At4g36700 corresponding to a late embryogenesis abundant protein of the RmlC-like cupin superfamily and the chloroplast ribosomal protein Rps7. Whereas cupin may be found due to a spurious interaction related to its high abundance in the young seedling, the presence of Rps7 may be due to the proximity of the PEP to the ribosome. Such a proximity is observed in bacteria and is referred to as transcription–translation coupling. The remaining intruders among the 24 highly abundant proteins belong to the family of histones, suggesting that some nucleosomes copurify with the PEP fraction. This contamination is likely due to nuclei associated with the chloroplast envelopes. All other detected proteins are in the background noise (low stoichiometry of the peptides) corresponding to low-abundant proteins and reflecting the high sensitivity of mass spectrometry. In electron microscopy, though, the contaminant proteins within the sample did not interfere with the structural analysis of the PEP complex, since the individual particles appeared homogenous enough for the calculation of its three-dimensional envelope (Figure 1c and Figure 2 below).

### 2.2. Patches of Specific Residues Are at the Surface of the PEP Catalytic Core

We hypothesized that the emergence of PAPs in the green lineage became essential for chloroplast biogenesis in angiosperms when all the PAPs acquired the capacity to bind to the core enzyme, hereby controlling its transcriptional activity in a “go/no go” switch that remains to be elucidated. It is then implying that surfaces of interactions on the core have evolved, possibly generating innovations (differences with ancestors) that are under selection pressure for conservation (Figure 3). To highlight the differences in the PEP core complex that could be evolutionarily associated with PAP interactions compared to eubacterial RNAPs, we performed a detailed sequence alignment analysis of the α, β, β’ and β” core subunits from various species chosen in the tree of the green lineage, as proposed by Finet et al. [36]. These sequences were found to be well-conserved within the green lineage (Figure 4 and Appendix A). The lowest sequence identity is observed when comparing *Physcomitrium* to other species, the sequence of the α subunit being the most divergent. Sequence conservation appears to be high in the domains of the β, β’ and β” subunits that bear the catalytic activity, while it is lower for the α subunits that are responsible for the assembly of the core [37]. Sequence comparisons with RNAPs from bacteria and cyanobacteria reveal that the regions that are essential for the transcription activity are conserved, and the bacterial β’ subunit can be aligned with the β’ and β” subunits of the PEP (Figure 4b, Appendix A).

Whereas the catalytic activity is carried by the β and β’ subunits in bRNAPs, it is supported in the PEP by the β, β’ and β” subunits. Unlike in *E. coli*, the β subunit of the PEP does not have the additional βi4, βi9 and βi11 domains [38]. However, the β” subunit of the PEP contains a long plant-specific insertion of several hundred residues between regions β’b8 and β’b9 that does not exist either in the β’ subunit from *E. coli* RNAP or in the β’ subunit from *T. thermophilus* RNAP (Figure 4b,c and Appendix A). The β” subunit of RNAP from angiosperms also lacks a part of the β’b10 region observed in the RNAP from *Nostoc*. Nevertheless, most of the strictly conserved residues described for the catalytic cores of RNAPs [39] are conserved in the PEP. The amino acid homologies were mapped on the *E. coli* 3D structure (Figure 3). Most of the variable sites in the PEP sequences are located at the surface of the catalytic core of the bRNAP, supporting the assumption that some of these innovations may be required for the interaction with PAPs (Figure 3a). The overall difference in amino acid functionalities, however, is rather limited outside of the β” large insertion that remains invisible in these representations (Figure 3b).

### 2.3. A Chloroplast Catalytic Core Surrounded with Nuclear-Encoded Proteins

We then investigated the 3D structure of the fully assembled PEP complex by using negative-staining electron microscopy. The overview images of the stained complexes displayed well-separated molecules of various shapes but very limited aggregation (Figure 2) and no disturbance by other complexes (such as nucleosomes). The homogeneity of the sample was probed by ab initio 2D classification of the individual complex images that revealed several well-defined 2D classes (Figure 2b). The overall shapes of the classes are multiple, but they are all consistent in sizes with dimensions varying between 150 and 280 Å. Some 2D classes of PEP displayed a more compact center, sometimes with a clear stain-filled pocket surrounded by several protrusions of various sizes (Figure 2b). From the particles isolated by 2D classification, a 3D map at 27.5 Å resolution could be determined (Figure 2c), which recapitulates the features seen in the 2D classes, such as the central cavity (depression) and the peripheral protrusions. The resolution was not sufficient, though, to confidently fit the catalytic core of the *E. coli* RNAP in the PEP-A envelope.

In order to obtain information about the relative position of the PEP subunits within the complex, we used a biochemical crosslinking coupled to MS. To this end, we treated the PEP-enriched fraction from two independent purifications (replicates 2 and 3 of the preparations used for the proteomic discovery) with Disuccinimidyl Dibutyric Urea (DSBU) before tryptic digestion and the MS analyses. This strategy allowed to reliably identify 39 interprotein dipeptides, 12 of which contained PEP core subunits or PAPs, suggesting a spatial proximity between these subunits within the PEP complex (Table 1 and Appendix A).

The core subunits were partly accessible to the DSBU treatment, since two dipeptides linking the β and β’ subunits were identified, suggesting that the associated PAPs do not cover the core completely but leave some gaps that allow the crosslinker molecules to access the core. Structure analyses of the RNAPs from *E. coli* (PDB entries: 3LU0 [38] and 6GH5 [40]) and *T. thermophilus* (PDB entry: 6ASG [41]) do not allow to model the dipeptides observed, suggesting that these regions in the PEP have different conformations despite their sequence conservation (Appendix A). PAP5 and FLN2 were found to both interact with the same peptide of the α subunit, indicating that PAP5 interacts with one monomer while FLN2 interacts with the second monomer (Table 1 and Figure 2c and Figure 5). A distinct region of PAP5 was found in close vicinity to the KNYQNER peptide of the β’ subunit (Table 1) that belongs to an insertion of conserved residues found only in angiosperms after the β’a12 domain (Figure 2c and Appendix A). This result supports the assumption that surface-localized residues that are not conserved between the catalytic cores of bRNAPs and PEP but conserved in plants have evolved towards the interactions with the PAPs (see below). We also found a PAP5-FLN2 dipeptide, suggesting that the α, β and β’ subunits PAP5 and FLN2 may form a structural cluster within the fully assembled PEP complex (Figure 5a). A second cluster appears to be formed by PAP1, PAP2 and PAP11/MurE-like for which dipeptides were also found (Table 1 and Figure 5a).

## 3. Discussion

The purification protocol used in this study allowed us to retrieve a stable PEP complex with a limited amount of contaminant proteins. The core subunits and previously described PEP-associated proteins are the most abundant proteins. The three MS-based proteomic characterizations of the *Sinapis alba* PEP fraction revealed the presence of FLN1 (PAP6) and FLN2, two fructokinase-like proteins whose gene deletion lead, respectively, to an albino phenotype or a delayed greening [43]. FLN2 is the paralogous protein of FLN1, and despite its fructokinase domain, sugar-phosphorylating activity remains to be detected [26]. They can form homodimers or heterodimers in vitro [44]. Characterization of the proximal proteins in the *S. alba* PEP fraction using XL-MS showed that FLN1 or FLN2 interact with the α subunit of the catalytic core. Based on the sequence, it is not possible to distinguish which FLN paralog binds to the α subunit due to the high sequence identity between FLN1 and FLN2 that display the same identified peptide sequence. The part of the α subunit observed in this interaction (GY(157)SLK(160)MSNNFEDR) is the same that also interacts with PAP5, involving Y157 and K160 in the dipeptide bond with PAP5 and FLN1/FLN2, respectively. Considering that the complex has a homogenous structure with correctly positioned partners, steric hindrance would not allow for two proteins with predicted different folds (PAP5 and PAP6/FLN1 or FLN2) to interact with the same region of the α subunit. The MS-based proteomic characterization of the *S. alba* PEP fraction also suggested that the α subunit is twice more abundant than the β subunit. Together, these observations are consistent with a stoichiometry of two α subunits per one β subunit [37] in the PEP core complex. Hence, this supports the assumption that the PEP core resembles that of bRNAPs. In the PEP, PAP5 and FLN2 form a cluster with the α, β and β’ subunits, suggesting that they can associate early during a de novo PEP-B-to-PEP-A transformation. XL-MS also revealed the presence of two other closely related proteins, PAP1 and PAP2. Both PAP1 and PAP2 possess pentatricopeptide repeats involved in RNA binding. Among the PAPs with predicted nucleic acid-binding domains, PAP1 possesses a SAP domain known for DNA or RNA binding, while PAP3 has a S1-like domain predicted to interact with RNA [18]. Since dipeptides between PAP1 and PAP2 are found, both proteins may sit on the PEP as a heterodimer, PAP1 also being involved in interactions with PAP11/MurE-like (Appendix A), and the three proteins form a second cluster containing the largest PAPs.

The presence of closely related proteins, such as PAP6 and FLN2 or the two superoxide dismutases PAP4 and PAP9, raises the question of the PEP subunit composition. Even if the 3D classifications did not reveal any significant variability in the 3D envelope, PEP heterogenous complexes could exist. Furthermore, the PEP complex of our preparations could contain additional subunits such as FLN2 or pTAC18, not detected previously in gel-based MS analyses. It remains open whether these subunits represent loosely or tightly associated PEP subunits. The initial discovery of pTAC18 in the TAC already placed this protein conceptually close to the PEP [35]. Further biochemical analyses associated with a high-resolution cryo-EM map of the PEP and new XL-MS experiments with other crosslinkers will likely resolve the question about the bona fide PEP subunit composition and the potential existence of stage-specific differences.

Indeed, the PEP envelope was calculated at a resolution that does not allow fitting of the map with homologous structures of the catalytic core or PAPs such as PAP9 [32] or high-confidence PAP models. However, the proposed fitting of the catalytic core of the *E. coli* RNAP (PDB entry: 3LU0) [38] revealed the remaining space for the subsequent positioning of the PAPs (Appendix A). It is noteworthy that further 3D classifications did not reveal any significant variability in the 3D envelope of the PEP, suggesting that the protrusions that we attribute to the PAPs are firmly associated with the catalytic core. Despite the recognition of some structural features such as the cleft and stalk, the overall shape of the *S. alba* active PEP envelope is different from that of RNAPs II and III (Appendix A). The use of novel algorithms such as AlphaFold [42] is still limited to predicting larger complexes such as PEP-A in particular to address the spatial organization of the PAPs with the PEP core enzyme.

A sequence comparison (Appendix A) shows that the four insertion regions characterized in *E. coli* RNAP [38] do not exist either in PEP or in the RNAP from *Nostoc*. The high sequence identity between the catalytic core of the bacteria and plastids suggests that the overall shape of the PEP core and the associated catalytic activity are conserved. The bacterial β’ subunit has likely been split into two subunits during evolution after the separation of the eubacteria and cyanobacteria branches, the latest uniquely sharing the β” subunit with the chloroplast [45]. The sequence alignment showed that the β’ and β” subunits of the PEP can be, respectively, aligned with the N-terminal and C-terminal parts of the β’ subunit from bRNAPs. In addition, a very long insertion in the β” subunit of plastids and cyanobacteria (Phe364-Ser1093 in *A. thaliana*) is not observed in the C-terminal part of the β’ subunit from bRNAPs. This insertion is located in the trigger loop region at the surface of the bRNAPs (Appendix A). With such a length, this region could be an additional domain in the PEP associated with oxygenic photosynthesis.

Sequence divergence with the *T. thermophilus* and *E. coli* RNAPs is mainly observed between residues located at the surface of the core complex. Since the nuclear-encoded PAPs seems to have appeared with the terrestrialization of the green lineage (first appearance in fresh water algae and mosses), it is likely that the evolution of novel cell types requested some control of the PEP catalytic core activity, providing the capacity to generate novel plastid types. The PAPs, acting as signaling components expressed after phytochrome activation in the nucleus of angiosperms, may have been required to control PEP activity by the nucleus in order to synchronize the transcription of the photosynthesis-associated nuclear genes (*PhANGs*) and photosynthesis-associated plastid genes (*PhAPGs*) for the proper building of the photosynthetic apparatus upon first illumination. Due to their dual localization, some of the PAPs such as PAP5/HEMERA [46] and PAP8 [47,48,49] provided a potential regulatory link between the nucleus and plastids in the expression of photosynthesis genes. It remains to be solved whether their nuclear or their plastid function evolved first.

In conclusion, this study opens the road for an in-depth structural description of the PEP complex responsible for the expression of photosynthesis-associated plastid genes. This complex possesses a well-defined structure with subunits that are specifically associated with the catalytic core, providing essential functions related to efficient transcription, post-transcriptional modifications and protections against the threats of photosynthesis reactions.

## 4. Materials and Methods

Chloroplast isolation: Six to seven-day-old *Sinapis alba* cotyledons were collected and homogenized using a blender with short pulses (3 × 3 s): 100 g approximately of fresh material in 200 mL homogenization buffer containing 50 mM HEPES-KOH, pH 8.0, 0.3 M sorbitol, 5 mM MgCl_2_, 2 mM EDTA and 0.3 mM DTT. The suspension obtained was then filtered through a 56-µm nylon mesh, then centrifuged 3 min at 6084.1× *g* at 4 °C. The pellet was resuspended in homogenization buffer and poured in a potter to remove all the chloroplast aggregates. The suspension was then loaded on a linear percoll gradient (35% percoll, 50 mM HEPES-KOH, pH 8.0, 0.3 M sorbitol, 5 mM MgCl_2_, 2 mM EDTA and 0.3 mM DTT) and centrifuged 50 min at 4696× *g*, 4 °C. The fractions containing the chloroplasts were then pooled, diluted in homogenization buffer and centrifuged 10 min at 4000× *g*, 4 °C to remove percoll. The pellet containing the chloroplasts was solubilized in the lysis buffer containing 50 mM Tris HCl, pH 7.6, 25% glycerol (*w/v*), 10 mM NaF, 4 mM EDTA, 1 mM DTT and 1% Triton X-100 (*w/v*) and poured in a potter for homogenization. The suspension was then centrifuged 1 h at 15,000× *g*, 4 °C and the supernatant frozen in liquid nitrogen and stored at −80 °C before using it to purify the PEP.

PEP purification: After thawing, the chloroplast lysate was mixed overnight at 4 °C with heparin resin equilibrated with 50 mM HEPES, pH 7.6, 10% (*w/v*) glycerol, 10 mM MgCl_2_, 80 mM (NH_4_)_2_SO_4_, 1 mM DTT and 0.1% (*w/v*) Triton X-100. The resin was extensively washed with 50 mM HEPES, pH 7.6, 10% (*w/v*) glycerol, 10 mM MgCl_2_, 80 mM (NH_4_)_2_SO_4_, 1 mM DTT and 0.1% Triton X-100 (*w/v*) before elution over 10 fractions of 1 mL with 50 mM HEPES, pH 7.6, 10% (*w/v*) glycerol, 10 mM MgCl_2_, 1.2 M (NH_4_)_2_SO_4_, 1 mM DTT and 0.1% Triton X-100 (*w/v*). The fractions were then subjected to SDS-PAGE and Western blot analyses with anti-PAP8 antibodies [47]. The fractions containing PAP8 and, therefore, the PEP were pooled; loaded on a 35–15% glycerol gradient (50 mM HEPES, pH 7.6, 35–15% (*w/v*) glycerol, 10 mM MgCl_2_ and 0.01% (*w/v*) Triton X-100) and centrifuged at 97,083× *g* on a SW55-Ti rotor (Beckmann Coulter) for 16 h at 4 °C.

The gradient was then analyzed using SDS-PAGE and Western blot. The fractions containing the PEP were pooled before the last step of purification or frozen in liquid nitrogen and stored at −80 °C. The pool containing the PEP was mixed overnight with Q-Sepharose resin (Amersham) pre-equilibrated in 50 mM HEPES, pH 7.6, 10% glycerol (*w/v*), 10 mM MgCl_2_ and 0.01% (*w/v*) Triton X-100. The complex was eluted using a 0–1 M NaCl gradient. The fractions containing the PEP were pooled and concentrated at 2000× *g* on a 100-kDa cutoff membrane. The purified PEP was then frozen in liquid nitrogen and kept at −80 °C before analyses.

Sequence alignments: Full-length coding sequences of the α, β, β’ and β” subunits were retrieved from Blastp. The protein sequences were aligned using Clustal Omega (https://www.ebi.ac.uk/Tools/msa/clustalo/ accessed on 1 January 2022) and then colored using the BOXSHADE server using default parameters. The domains of the α, β, β’ and β” subunits of the PEP were assigned based on those described [37,39].

MS-based proteomic analyses: Three PEP preparations from independently grown plant batches were analyzed. For this, purified PEP from chloroplasts was solubilized in Laemmli buffer and stacked in the top of a 4–12% NuPAGE gel (Invitrogen). After staining with R-250 Coomassie Blue (Bio-Rad), the proteins were digested in gel using trypsin (modified sequencing purity, Promega), as previously described [49]. The resulting peptides were analyzed by online nano-liquid chromatography coupled with MS/MS (Ultimate 3000 RSLCnano and Q-Exactive Plus, Thermo Fisher Scientific) using a 140-min gradient. For this purpose, the peptides were sampled on a precolumn (300 μm × 5 mm PepMap C18, Thermo Scientific) and separated in a 75 μm × 250 mm C18 column (Reprosil-Pur 120 C18-AQ, 1.9 μm, Dr. Maisch). The MS and MS/MS data were acquired by Xcalibur (Thermo Fisher Scientific). Peptides and proteins were identified by Mascot (version 2.7, Matrix Science) through concomitant searches against the NCBI database (*Sinapis alba* strain: S2 GC0560-79 (white mustard) taxonomy, BioProject PRJNA214277, July 2020 download), the UniProt database (*Sinapis alba* taxonomy, February 2021 download), a homemade database containing the sequences of classical contaminant proteins found in proteomic analyses (human keratins, trypsin, etc.) and the corresponding reversed databases. Trypsin/P was chosen as the enzyme, and two missed cleavages were allowed. Precursor and fragment mass error tolerances were set, respectively, at 10 and 20 ppm. Peptide modifications allowed during the search were: Carbamidomethyl (C, fixed), Acetyl (Protein N-term, variable) and Oxidation (M, variable). The Proline software [50] was used for the compilation, grouping and filtering of the results (conservation of rank 1 peptides, peptide length ≥ 6 amino acids, peptide score ≥ 25, allowing to reach a false discovery rate of the peptide spectrum match identifications < 1%, as calculated on the peptide spectrum match scores by employing the reverse database strategy and the minimum of one specific peptide per identified protein group). Proline was then used to perform a MS1 label-free quantification of the identified protein groups based on razor and specific peptides. Intensity-based absolute quantification (iBAQ) [34] values were calculated from MS1 intensities of razor and specific peptides. The iBAQ values of each protein were normalized by the sum of the iBAQ values of all the quantified proteins in each sample before summing the values of the three replicates to generate the final iBAQ value. The gene names for the identified proteins were annotated after the Blastp search for the *A. thaliana* proteome.

Crosslinking coupled to MS analyses: A few micrograms of two PEP preparations used for mass spectrometry-based proteomic analyses (replicates 2 and 3) were crosslinked during 1 h at room temperature using 100 µM of DSBU in HEPES buffer, pH 7.8. To quench the crosslinking reaction, one microliter of 1 M ammonium bicarbonate was added and the sample incubated for 15 min at room temperature. To reduce disulfide bonds, 100 mM DTT solution was added to obtain a final concentration of 3.5 mM, and the mixture was incubated at 56 °C for 30 min in a ThermoMixer. For the alkylation of cysteines, 50 mM IAA solution was added to a final concentration of 8 mM, and the mixture was incubated at room temperature in the dark for 20 min. Freshly prepared trypsin solution to an enzyme/protein ratio of ~1:50 was added, and the digestion was performed overnight at 37 °C. To quench the enzymatic digestion, a final TFA concentration of 1% (*v/v*) was added. Micro-spin columns (Harvard Apparatus) were then used to desalt the samples using 5% ACN, 0.1% TFA as the washing solution and 75% ACN, 0.1% TFA as the elution buffer.

The resulting peptides were analyzed by online nano-liquid chromatography coupled with MS/MS (Ultimate 3000 RSLCnano and Orbitrap Exploris 480 for replicate 2 and Q-Exactive HF for replicate 3, Thermo Fisher Scientific). Peptides were sampled on a precolumn (300 μm × 5 mm PepMap C18, Thermo Scientific) and separated using a Pharmafluidics μPAC™ column of 200 cm in length (with a pillar array backbone at an interpillar distance of 2.5 μm) using a 240-min method. Data were acquired in the data-dependent MS/MS mode with stepped higher-energy collision-induced dissociation (HCD) and normalized collision energies (20%, 25% and 35% for Orbitrap Exploris 480 and 22%, 27% and 30% for Q-Exactive HF).

Data analysis was conducted using MeroX 2.0 [51]. The following settings were applied: proteolytic cleavage: C-ter at Lys and Arg with 3 missed cleavages allowed, peptide length 4–30 amino acids, fixed modification: alkylation of Cys by IAA and variable modification: oxidation of Met, crosslinker: DSBU with specificity towards Lys, Ser, Thr, Tyr and N-ter for site 1 and 2, analysis mode: RISEUP mode, maximum missing ions: 2, precursor mass accuracy: 10 ppm, product ion mass accuracy: 30 ppm, signal-to-noise ratio: 2, precursor mass correction activated, pre-score cutoff at 10% intensity, FDR cut-off: 1% and minimum score cut-off: 30. Crosslinks identified in the two replicates were then combined using Merox.

Negative staining electron microscopy: Ten microliters of PEP were added to a glow discharge grid coated with a carbon-supporting film for 3 min, and the grid was stained with fifty microliters of Sodium Silico Tungstate (SST) (1% (*w/v*) in distilled water (pH 7–7.5)) for 2 min. The excess solution was soaked by a filter paper, and the grid was air-dried. The images were taken at 30,000 magnification (2.2 Å/pixel) under low-dose conditions (<10 e-/Å^2^) with defocus values between −1.2 and −2.5 μm on a Tecnai 12 (Thermo Fischer Scientific) LaB6 electron microscope operating at 120 kV using a Gatan Orius 1000 CCD camera.

Determination of the PEP envelope: The image processing was entirely done in RELION [52]. The CTF parameters of each micrograph were determined with CTFFIND4 [53], and the particles were auto-picked in RELION with the Laplacian of the Gaussian option. Two-dimensional classification was then performed in 50 classes using a 350 Å mask diameter that resulted in the selection of 17,567 particles. The latter were then used to create an ab initio model (C1 symmetry and 300 Å mask diameter) that was then used to calculate a 3D map (C1 symmetry and 320 Å mask diameter) at 27.5 Å resolution (at FSC = 0.143) (Appendix A).

## Figures and Tables

**Figure 2 ijms-23-09922-f002:**
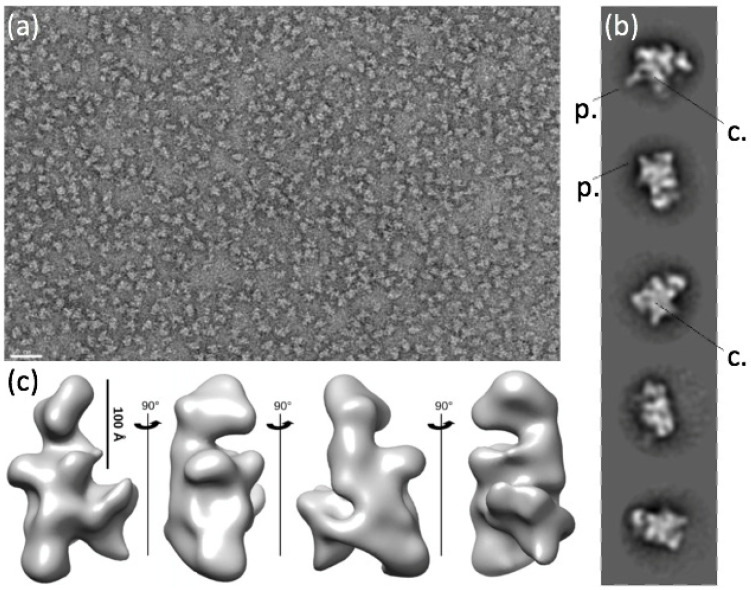
Negative-staining electron microscopy and 3D envelope of the PEP-A complex. (**a**) Overview image of the grid after negative staining. Note the homogeneity of the sample and the lack of other protein complexes. The white scale bar represents 50 nm. (**b**) Two-dimensional classes of PEP. (**c**) Three-dimensional envelope of PEP at 27.5 Å resolution calculated from 17,567 particles.

**Figure 3 ijms-23-09922-f003:**
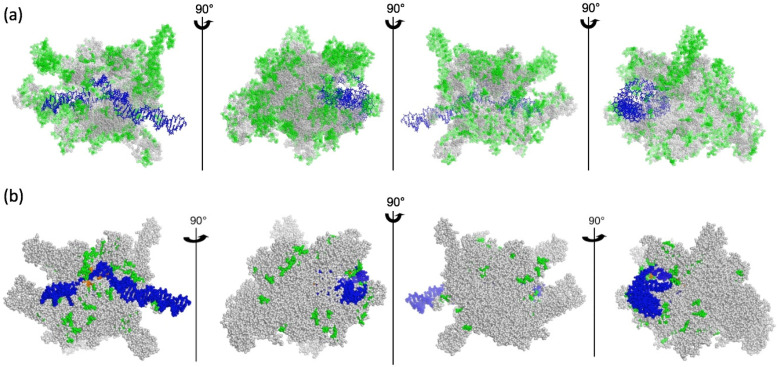
Mapping variable sites of the core subunits. View of the *E. coli* RNAP (PDB entry: 6GH5) without the ω subunit and the σ54 factor. The double-stranded DNA is colored in blue. The core subunits are drawn in spheres. (**a**) Mapping the variable sites as homologous in grey and nonhomologous or gaps in green. (**b**) Mapping only amino acid functional differences between bRNAP and PEP, as given in the sequence alignments (Appendix A) The residues colored in green and orange are those displaying a strong modification of functional groups for at least 3 consecutive amino acids.

**Figure 4 ijms-23-09922-f004:**
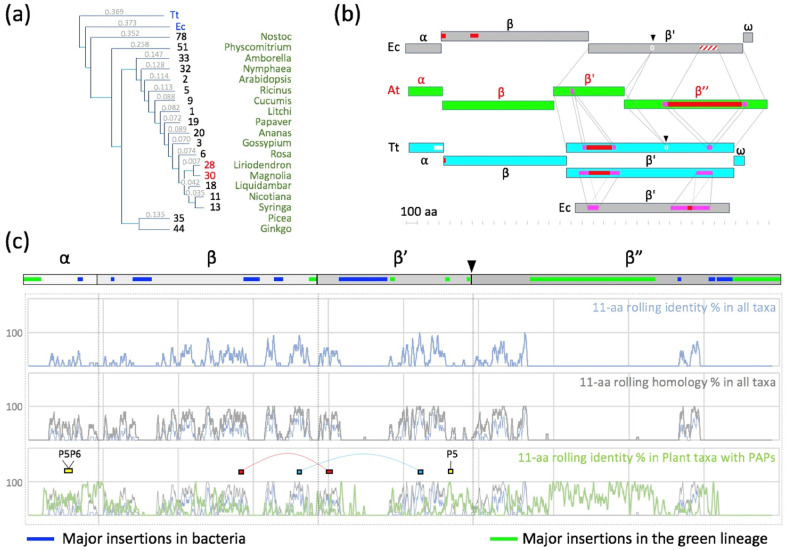
Phylogeny and sequence alignments of the core subunits. (**a**) Phylogram obtained with the Clustal Omega multi-alignment algorithm. Branch length presented as a cladogram. Major taxa included from the collection presented in the Appendix A. A major incongruence from the angiosperm phylogeny tree (version IV: http://www.mobot.org accessed on 1 January 2022) is noted for Magnoliales and likely due to the study of chloroplast genes with cytoplasmic inheritance. (**b**) Schematic representation of the sequence context of *E. coli* (Ec), *T. thermophilus* (Tt) and *A. thaliana* (At) RNAP or PEP subunits as the output of a dot plot analysis performed using dotmatcher (https://www.bioinformatics.nl/cgi-bin/emboss/dotmatcher accessed on 1 January 2022). Insertions are represented in red or dashed red, with the duplicated area in pink. The splits of the bacterial β’ in the PEP β’ and β” are presented with a light-grey circle and a black triangle separating the shared regions. (**c**) Global alignment represented as the 11-aa rolling identity (blue) or homology (grey) percentages calculated for all taxa. In green is the 11-aa rolling identity percentage calculated in a subset of taxa corresponding to plants with detected PAPs (green). The black triangle is the evolutionary split of the rpoC gene in the rpoC1 and rpoC2 genes in the cyanobacteria. Red and blue rectangles represent dipeptides between β-β’, while yellow rectangles represent interacting peptides with PAPs, as found in the XL-MS analysis (see below).

**Figure 5 ijms-23-09922-f005:**
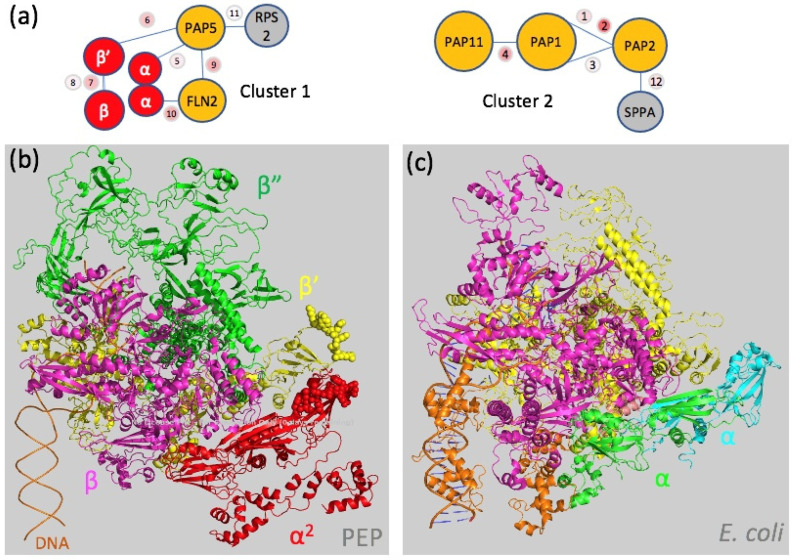
Mapping protein interactions on the core complex. (**a**) Protein clusters determined from the XL-MS analysis (Table 1) are schematically presented with the link and scores; grey bubbles correspond to the protein not belonging to the PEP-A purified complex: RPS2, Ribosomal Protein S2; SPPA, light-inducible chloroplast protease complex associated with thylakoid membranes. Cluster 1 composed of the PAP5, FLN2, α and β’ subunits. Cluster 2 composed of PAP1, 2 and 11. (**b**) Model of the PEP core complex from *A. thaliana* built from the α, β, β’ and β” subunits modelized using AlphaFold [42] and superimposed onto the *E. coli* RNAP catalytic core and colored as follows: α subunit in red, β subunit in pink, β’ subunit in yellow and β” in green. The van der Waals spheres display the peptides of the α and β’ subunits that are nearby to PAP5 and FLN2 (Table 1). (**c**) View of the catalytic core from the *E. coli* RNAP (PDB entry: 6GH5 [40]).

**Table 1 ijms-23-09922-t001:** Characterization of the proximal proteins in the *S. alba* PEP fraction using crosslinking-MS. Selection of the 12 best hetero-dipeptides is presented with the corresponding protein partners, crosslink score, peptide sequences, position and crosslinked amino acid with a relative position to the peptide. The overall dipeptides are given in Appendix A.

#	Protein 1 Names	Protein 2 Names	xLinkScore	Peptide 1	From	To	aa 1	Peptide 2	From	To	aa 2
1	PAP1/pTac3	PAP2/pTac2	72.00	[KELGAGQRPLPETMIALVR]	131	149	K1	[GQLEKSSAAR]	753	762	K5
2	PAP1/pTac3	PAP2/pTac2	194.61	[KELGAGQRPLPETMIALVR]	131	149	K1	[GQLEKSSAAR]	753	762	K5
3	PAP1/pTac3	PAP2/pTac2	49.10	[ENEDSSSFGSSEAVSALER]	50	68	S15	[GQLEKSSAAR]	753	762	S6
4	MURE	PAP1/pTac3	133.68	[ELKPR]	608	612	K3	[VQKAR]	564	568	K3
5	SaRpoA	PAP5/PTAC12	57.62	[GYSLKMSNNFEDR]	156	168	Y2	[IKRDPLAMR]	365	373	K2
6	PAP5/PTAC12	SaRpoC1	99.25	[KLGRPHPFIDPTK]	208	220	K1	[KNYQNER]	683	689	K1
7	SaRpoC1	SaRpoB	108.78	[IFGPIKSGIBABGNYR]	60	75	Y15	[LTPQVAKESSYAPEDR]	733	748	K7
8	SaRpoC1	SaRpoB	52.00	[FRETLLGKR]	489	497	K8	[SKQGGQR]	969	975	S1
9	PAP6/FLN1; FLN2	PAP5/PTAC12	89.14	[KLELVGSMGEDDDSS}	602	617	K1	[NWSVLKSTPELR]	481	492	K6
10	PAP6/FLN1; FLN2	SaRpoA	121.48	[MLTVQPDLMNDKGYLER]	505	521	Y14	[GYSLKMSNNFEDR]	156	168	K5
11	PAP5/PTAC12	RPS2A; RPS2B	39.00	[APQPAGESSSFPSYGKNPGSR]	128	148	S20	[EVATAIR]	137	143	T4
12	PAP2/pTac2	SPPA	67.62	[GGLFKESEVILSR]	503	515	S7	[GQISDQLKSR]	135	144	K8

## Data Availability

The MS and crosslinking coupled to the MS data and were deposited to the ProteomeXchange Consortium via the PRIDE [54] partner repository with the dataset identifiers PXD032738 and PXD032739, respectively. Each spectrum corresponding to an interprotein link with the best scores were manually checked. The eM data were deposited into the EMDB with accession code EMD-14571.

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
