# Peer review of "Three-Dimensional Envelope and Subunit Interactions of the Plastid-Encoded RNA Polymerase from Sinapis alba"

_ijms, 2022, doi:10.3390/ijms23179922_

Round 1
Reviewer 1 Report
Plastid-encoded RNA polymerase (PEP) is critical machinery for the transcription process in plant plastids. In this manuscript, the authors resolved its 3D structure and performed an in-depth analysis of its protein components both in the core and the envelope. This study is very helpful for a good understanding of the PEP structure and catalytical/regulatory mechanisms. The manuscript is clealry written and well organized. The only suggestion from this reviewer, which is probably largely beyond the scope of this current manuscript, is to incorporate the sequence and maybe also structural information from unicellular red algae, such as Cyanidioschyzon merolae, in Figure 2. Some of their plastid genes have not been transferred to the nuclear genome.
Author Response
This is a good suggestion but, as said by the reviewer himself, it adds a level of complexity that is far beyond the scope of this manuscript. The catalytic core of the red algae Cyanidioschyzon merolae is composed of only the α, β, and β’ subunits unlike the chosen modern cyanobacteria and green chloroplasts. Since red algae do not possess PAPs, we prefer to exclude them from this study.
It could serve as an outgroup for rooting phylogenetic trees though, but this would trigger an unnecessary discussion about the monophyletic origin of the chloroplast.
Reviewer 2 Report
The authors have investigated the molecular structures and potential functions of the the plastid-encoded RNA polymerase (PEP) and associated proteins (PAPs) based on Sinapis alba. I have the following comments for the authors to consider if a revision is requested by the editor.
Abstract: The Abstract needs to be shortened, in particular, lines 18-25, the lengthy background. More main results need to be included. The conclusions, including the significance of this study, should be explicitly defined.
Introduction: I believe that the authors provided sufficient background but missing the specific goals of this study and its significance, i.e., conclusions withdrawn based on available data. This information may be provided in the last paragraph of the Introduction.
Results: I believe that the authors have presented the results with appropriate figures and tables. There are several minor revisions that I would like to indicate for the authors to consider if a revision is requested by the editor.
Figure 2. The title of Figure 2 “Phylogeny of the core subunits” is not sufficient because it lacks the summary of (b) and (c) on this figure.
Figure 5. The overall summary title of Figure 5 is missing.
M&M and Discussion: I believe that the authors explained the methodology well and discussed the results well in relation to relevant studies. However, again, some conclusive remarks are missing, as I have indicated above. This problem would be solved by providing a separate section “Conclusions” somewhere at the end of the manuscript.
Author Response
Abstract: The Abstract needs to be shortened, in particular, lines 18-25, the lengthy background. More main results need to be included. The conclusions, including the significance of this study, should be explicitly defined.
The abstract has been shortened and modified as suggested by the reviewer.
Introduction: I believe that the authors provided sufficient background but missing the specific goals of this study and its significance, i.e., conclusions withdrawn based on available data. This information may be provided in the last paragraph of the Introduction.
We modified the introduction.
Lines 100-103: So, understanding chloroplast biogenesis associated with photosynthesis in angiosperms requires to study the nuclear-encoded PAPs that, added to PEP, regulate gene expression while protecting the machinery from
the threatening reactions of photosynthesis.
Lines 135 to 139: information have been added
Results: I believe that the authors have presented the results with appropriate figures and tables. There are several minor revisions that I would like to indicate for the authors to consider if a revision is requested by the editor.
Figure 2. The title of Figure 2 “Phylogeny of the core subunits” is not sufficient because it lacks the summary of (b) and (c) on this figure.
Figure 2. Phylogeny of the core subunits.
Figure 2. Phylogeny and sequence alignments of the core subunits.
Figure 5. The overall summary title of Figure 5 is missing.
Figure 5. (a)
Figure 5. Mapping protein interactions on the core complex (a)
M&M and Discussion: I believe that the authors explained the methodology well and discussed the results well in relation to relevant studies. However, again, some conclusive remarks are missing, as I have indicated above. This problem would be solved by providing a separate section “Conclusions” somewhere at the end of the manuscript.
We added (Lines 369-374): In conclusion, this study opens the road for in-depth structural description of the PEP complex responsible for the expression of photosynthesis-associated plastid genes. This complex possesses a well-defined structure with subunits that are specifically associated to the catalytic core, providing essential functions related to efficient transcription, post-transcriptional modifications and protections against the threats of photosynthesis reactions.
Round 2
Reviewer 2 Report
I appreciate very much the efforts that the authors have devoted to improving their manuscript. I have no more questions.